# *Crocus sativus* L. Extract (Saffron) Effectively Reduces Arthritic and Inflammatory Parameters in Monotherapy and in Combination with Methotrexate in Adjuvant Arthritis

**DOI:** 10.3390/nu15194108

**Published:** 2023-09-22

**Authors:** Martin Chrastina, František Dráfi, Katarína Pružinská, Silvester Poništ, Kevine Silihe Kamga, Sasan Khademnematolahi, František Bilka, Peter Novák, Ľudmila Pašková, Katarína Bauerová

**Affiliations:** 1Institute of Experimental Pharmacology and Toxicology, Centre of Experimental Medicine SAS, 841 04 Bratislava, Slovakia; martin.chrastina@savba.sk (M.C.); katarina.pruzinska@savba.sk (K.P.); exfasipo@savba.sk (S.P.); kamkevine@yahoo.com (K.S.K.); sasan.khademnematolahi@savba.sk (S.K.); 2Jessenius Faculty of Medicine in Martin, Comenius University in Bratislava, Malá Hora 10701/4A, 036 01 Martin, Slovakia; 3Faculty of Science, University of Yaoundé 1, Yaoundé P.O. Box 812, Cameroon; 4Faculty of Medicine and Biomedical Sciences, University of Yaoundé 1, Yaoundé P.O. Box 1364, Cameroon; 5Faculty of Natural Sciences, Comenius University in Bratislava, Ilkovičova 6, 842 15 Bratislava, Slovakia; 6Faculty of Pharmacy, Comenius University in Bratislava, Odbojárov 10, 832 32 Bratislava, Slovakia; bilka@fpharm.uniba.sk (F.B.); novak114@uniba.sk (P.N.); paskova@fpharm.uniba.sk (Ľ.P.)

**Keywords:** saffron, methotrexate, arthritis, combination therapy, interleukins, MMP-9, GGT, CRP, mRNA expression

## Abstract

Rheumatoid arthritis (RA), an autoimmune disease, is characterized by inflammation that affects not only the liver but also other organs and the musculoskeletal system. The standard therapy for RA is methotrexate (MTX), which has safety limitations. The extract from *Crocus sativus* L. (saffron—SF) is also known for its anti-inflammatory effects. Therefore, we decided to investigate the potential benefit of SF in monotherapy via two doses (SF1—25 mg/kg of b.w.; SF2—50 mg/kg of b.w.) and in combination with MTX (0.3 mg/kg of b.w., twice a week) using adjuvant arthritis in rats. To evaluate these therapeutic settings, we used biometric, immunological, and biochemical parameters, as well as the relative gene expression of the mRNA in the liver. Our results showed a statistically significant increase in the experimental animals’ body weight and the arthritic score (AS) on day 14 for monotherapy with SF1 and SF2. The change of hind paw volume (CHPV) was significant only for SF2 monotherapy on the 14th day of the experiment. A combination of SF1 and SF2 with MTX significantly modulated all the biometric parameters during the experimental period. Additionally, AS and CHPV improved considerably compared to MTX monotherapy on day 21. Furthermore, all monotherapies and combination therapies were significant for the biochemical parameter γ-glutamyl transferase (GGT) in the joint. GGT activity in the spleen was less pronounced; only MTX in combination with SF1 significantly modified this parameter. The higher dose of SF monotherapy (SF2) was similarly significant with respect to immunological parameters, such as plasmatic IL-17A, IL-1β, and MMP-9 on day 21. The combination of both doses of SF with MTX significantly improved these immunological parameters, except for C-reactive protein (CRP), which was influenced only by the higher dose of SF2 in combination with MTX in plasma at the end of the experiment. A different effect was found for the relative expression of CD36 mRNA, where only SF1 significantly decreased gene expression in the liver. However, the relative gene mRNA expression of IL-1β in the liver was significantly reduced by the SF monotherapies and the combination of both SF doses with MTX. Our findings showed SF’s partial antiarthritic and anti-inflammatory potential in monotherapy, but the effect was stronger in combination with MTX.

## 1. Introduction

Autoimmune disorders are defined by immunological abnormalities that result in aberrant B-lymphocyte and T-lymphocyte cell reactivity to the typical host components because of poor regulation, genetic predisposition, and environmental factors [1,2]. In other words, it is a pathological condition in which the immune system is malfunctioning, destroying its cells because it recognizes them as foreign [3,4]. Rheumatoid arthritis (RA) is the most prevalent of these autoimmune disorders, with a prevalence of 0.5 to 1% worldwide. This means that RA affects about 70 million individuals, 80% of whom are women, and it has extra-articular manifestations as well [5,6]. Upon recognition of pathogen-associated molecular patterns (PAMPs) from microbial sources, the pattern-recognition receptors present in the cells involved in the innate immune response are activated. This starts the local inflammatory processes. This inflammatory reaction engages adaptive response cells, mainly B-lymphocyte and T-lymphocyte, generating antibodies against the molecules themselves. The responses of lymphocytes are mediated by antigen-specific clones in the joint [7]. Throughout the pathogenesis and even the pre-articular phase, cytokines are involved and promote autoimmunity by mediating chronic inflammation of synovium, followed by the destruction of adjacent joint tissue [8]. RA develops when the immune system of the body acts inappropriately, attacking healthy tissue and triggering inflammation. This inflammation causes pain and swelling in the joints and may eventually result in irreversible joint damage and severe disabilities [9]. Long-lasting diarthrodial joint inflammation in RA causes symmetric polyarthritis, synovial membrane thickening, progressive joint degeneration, cartilage and bone degradation, and deformity [10]. Despite predominantly affecting the synovial lining of the joints, RA can also have an impact on the heart, lungs, and blood vessels. 

Non-steroidal anti-inflammatory drugs (NSAIDs) and glucocorticoids, as well as disease-modifying antirheumatic drugs (DMARDs) such as methotrexate and hydroxychloroquine, are currently used as pharmacologic therapies for the treatment of RA. These drugs not only improve symptoms but also slow clinical and radiographic progression [11]. These treatments, however, are associated with a number of negative side effects, comprising biological therapy or intense immunosuppressive treatment, which seriously harms the organism’s immunological balance and raises the risk of infections [12].

These factors have created a pressing need for alternate immunomodulatory strategies that could reduce the negative effects of immunosuppressive medication on cellular and humoral immunity. This strategies has been recognised as an unmet medical need as well. Medicinal plants constitute immunosuppressive agents due to their inhibitory activity on the immune system’s cellular and humoral immune response [13]. Equally, they have been widely used for the management of many diseases as they are the primary source of drugs, easily accessible and rich in bioactive components. Due to their capacity to either down-regulate the actions of immune cells or limit the generation of inflammatory cytokines, their phytoconstituents have become safer immunomodulatory therapies and have been utilized over time to maintain immunological homeostasis [14]. A natural immunosuppressant’s primary mode of action depends on its antagonistic activity on oxidative stressors, which, in turn, cause the generation of free radicals that target cytokines and autoantibodies [15]. Many studies on the treatment of RA have thus been carried out with natural plant extracts, compounds, and mixtures of extracts and tested in clinical settings. They have revealed a number of biologically active substances with a broad application and the ability to down-regulate the potential side effects of the available drugs [16]. Among these extracts are *Cinnamomum cassia Presl*, *Ligusticum chuanxiong Hort*, *Aconitum kusnezoffii*, and *Rhodiola rosea*, which have ability to modify immunity and could lower the levels of inflammatory markers—e.g., interleukin (IL)-1β, IL-1, IL-6, prostaglandin E2 (PGE2), and transforming growth factor-α (TGF-α)—in plasma and in the synovial fluid [17,18,19,20,21]. All these data demonstrate the extracts’ antioxidant, anti-inflammatory, and immunomodulatory efficacy in proper RA management and in alleviating RA pain and symptoms.

A member of the *Iridaceae* family, *Crocus sativus* L. (saffron) is mainly cultivated in Iran, India, Greece, Morocco, and Spain. Saffron is a flowering plant with valuable nutrients and a broad spectrum of applications in alternative treatments for numerous diseases [22].

More than 150 volatile and non-volatile substances, including proteins, amino acids, polysaccharides, minerals, vitamins, pigments, polyphenols, alkaloids, saponins, terpenes, and their esters were found in this extract when subjected to phytochemical analysis. Safranal, crocin, crocetin, and picrocrocin are among the main bioactive substances [23,24,25].

These main constituents (e.g., crocin, crocetin, and safranal) have shown antioxidant, anti-inflammatory, anxiolytic, antidepressant, antihypertensive, hypolipidemic, analgesic, anticancer, anticonvulsant, and antinociceptive properties [26,27,28,29]. The potential of *C. sativus* L. to control inflammatory mediators, humoral immunity, and cell-mediated immune responses also allows it to operate as an anti-inflammatory and immunomodulatory agent, enhancing its antioxidant capacity [24]. In experimentally induced osteoarthritis, *C. sativus* L. was able to significantly decrease osteoarthritis-associated joint histological manifestations and decrease the level of pro-inflammatory immune cell subtypes after a 30-day treatment in Balb/c mice [22]. In light of this, the current investigation examined the antiarthritic activity of *Crocus sativus* L. both alone and in combination with the commonly prescribed medicine, methotrexate, in a Lewis rat model of adjuvant arthritis.

## 2. Materials and Methods

### 2.1. Animals for the Experiment: Housing, Environmental, and 3Rs Principles

The Department of Toxicology and Laboratory Animal Breeding Farm, Centre of Experimental Medicine, SAS, Dobrá Voda, Slovak Republic (permission No.: SKCH24016), provided the experimental animals for purchase. Five-week-old Lewis male rats were quarantined for seven days following arrival. The animals were kept in conventional conditions, with a 12 h/12 h light/dark cycle, a humidity level of 55%, and a temperature range from 21 °C to 24 °C. They also had unlimited access to a standard feed and tap water. The housing for the animals is in compliance with European Union Convention for the Protection of Vertebrate Animals Used for Experimental and Other Purposes—European Treaty Series—No. 123 [30]. The Ethics Committee at the Institute of Experimental Pharmacology and Toxicology, Centre of Experimental Medicine SAS in Bratislava, Slovakia, gave its approval to the experiment’s protocol (Protocol No.: SK UCH 04018) and the State Veterinary and Food Administration of the Slovak Republic, Bratislava (Document No.: 3144/16-221/3). The animals were sacrificed on the final day of the experiment under deep anaesthesia (a combination of Xylariem^®^ and Zoletil^®^). The EU guidelines for the treatment of experimental animals have been strictly adhered to. We have implemented the 3Rs (replacement, reduction, and refinement) philosophy in accordance with Directive 2010/63/EU [31]. The application of the 3Rs is also in accord with the scientific guidance of the European and International Cooperation on Harmonisation of Technical Requirements levels [32]. In the same way, we conducted this in vivo experiment according to all the current scientific principles outlined above.

### 2.2. Induction of AA in Male Lewis Rats 

Adjuvant arthritis (AA) is a well-known model of inflammation [33] that is frequently used at our institute [21,34]. A 0.1 mL suspension of heat-killed *Mycobacterium butyricum* (Difco, Detroit, MI, USA) with incomplete Freund’s adjuvants (Thermo Fisher Scientific, Waltham, MA, USA) at a dosage of 12 mg/mL was injected into experimental rats weighing 160–180 g. As previously mentioned [21,34,35], this suspension was injected intradermally at the rat’s tail base as a single dose. Each substance tested was given *per os* via a gastric tube, and the dose was specifically calculated based on the body weight (b.w.) of each animal. Every day before the tested substances were administered, the b.w. was measured.

### 2.3. Experimental Design of AA and Treatments 

After seven days of compulsory quarantine, the animals were randomized to seven experimental groups—1. HC; 2. AA; 3. AA-MTX; 4. AA-SF1; 5. AA-SF2; 6. AA-SF1 + M; and 7. AA-SF2 + M—and received treatment according to the group (Table 1). Healthy controls and untreated AA animals received vehiculum daily *per os* (tap water). All treated AA groups were given the experimental treatment according to the design (Table 1). Two AA groups also received methotrexate (MTX; EBEWE, Unterach am Attersee, Austria) in combination therapy. MTX was used at a sub-therapeutic dose of 0.3 mg/kg (two times per week) via concomitant administration of the experimental compound, i.e., the saffron extract in two doses applied daily (Table 1). The subtherapeutic dose MTX was applied in our investigation of the potential therapeutic effect on the experimental substances tested.

### 2.4. Evaluation of Experimental AA

Following the prior protocols with additional substances examined, for the evaluation of the arthritis, the biometric parameters were assessed [21,35].

The volume of the animals’ hind paw joints was measured on days 14 and 21 following the immunization. Change in hind paw volume (CHPV) was determined using a water plethysmometer (UGO BASILE, Comerio-Varese, Italy) and presented as the average elevation of percentage, i.e., the percent of the hind paw volume of each rat, compared to HPV measured on day 1. According to the following formula, the HPV on the chosen day was divided by the HPV on day 1, then represented as a percentage as follows: ([Day n]/[Day 1]) × 100 − 100 = value [%].

To ensure precise dosing, the animals’ body weight was monitored every day. The following formula was used to determine the changes in body weight (ChBW) on days 14 and 21: [Day n] − [Day 1] = value [g].

According to previous protocols with other substances tested, the arthritic score (AS) was calculated for each animal using the total score of the hind paw volume (in millilitres; maximum points: 8) plus the forelimb paw diameter (in millimetres; maximum points: 5) plus the diameter of the scab at the site of *Mycobacterium butyricum* application, measured parallel to the spinal column (in millimetres; maximum points: 5) [36,37].

### 2.5. Preparation of the Saffron Extract

The extract of saffron (*Crocus sativus* L.) was prepared from saffron stigmas grown in Bilgya village on the Absheron peninsula in Azerbaijan. Prof. Ulduz Hashimova of the Garayev Institute of Physiology of the Azerbaijan National Academy of Sciences, Azerbaijan, generously donated a saffron stigma alcohol extract for testing in AA and experimental osteoarthritis studies. The extract was developed via the following method: 5 g of dry, shredded saffron stigmas were extracted with 75% ethyl alcohol and matured for two days in the cold while being stirred continuously with a magnetic stirrer; the alcohol was distilled after the residue was filtered, rinsed with 75% alcohol, and filtered once more; the subsequently produced liquid was vacuum-extracted and concentrated as a dry residue between 40 and 50 °C. A viscous resinous material had a yield of the active ingredient in the extract which was almost 56% higher than that of the feed stock. Prior to the NMR and HPLC investigations, a portion of the lyophilized extract was kept at −20 °C. This method was first described by Boneva et al. (2023), as well as the following NMR and HPLC analyses [22]. Experimental animals received saffron stigma extract dissolved in water. 

The linearity and sensitivity of the HPLC method employed to identify and measure picrocrocin, crocin 1, and crocin 2 were validated. Dried *Crocus sativus* L. stigmas weighing 0.9617 g were extracted three times, yielding a 0.5516 g (57.36%) extract yield. Picrocrocin, crocin 1, and crocin 2 concentrations in the *Crocus sativus* L. extract were 1.82 ± 0.04 mg/g extract, 0.60 ± 0.07 mg/g extract, and 1.99 ± 0.04 mg/g extract, respectively, according to the HPLC measurement. Safranal was either undetectable in the examined extract or its concentration was below the measurement threshold.

The saffron extract used in our AA experiment was courtesy of our foreign partner, Assoc. Prof. Andrey Tchorbanov from the Stephan Angeloff Institute of Microbiology (SAIM), Bulgarian Academy of Sciences (BAS), Sofia. Assoc. Prof. A. Tchorbanov was, together with Dr. Katarina Bauerova, DSc., the principal investigator of the common bilateral project between BAS and the Slovak Academy of Science (SAS). The name of this common project is “BAS-SAS 2018/2021: The anti-inflammatory effects of astaxanthin, sulforaphane, and *Crocus sativus* L. extract evaluated in two rodent models of age-related diseases”. The HPLC and NMR and analyses were fully conducted by a team of Boneva and described by the same author (Boneva et al., 2023) in detail in a recent manuscript [22].

### 2.6. Plasma Samples Preparation and Evaluation of IL-17A, IL-1β, CRP, and MMP-9

According to our prior protocol [34,38], blood was collected from the retroorbital sinus of the rat’s eye on day 14 and centrifuged, followed by plasma storage at −70 °C. On the 21st day of the experiment, the tested animals were sacrificed under general deep anaesthesia. Following the manufacturer’s instructions, an enzyme-linked immunosorbent assay kit (R&D Systems, Minneapolis, MN, USA) was used to measure the concentrations of C-reactive protein, matrix metalloproteinase-9 (MMP9), IL-17A, and IL-1β in plasma samples.

### 2.7. The Activity of Gamma-Glutamyl Transferase in the Spleen and Hind Paw Joint Tissue

As in our earlier experiment [34], we utilized the method of Orlowski and Meister (1970) [39], modified by Ondrejickova et al. (1993) [40], to evaluate the activity of gamma-glutamyl transferase (GGT) on the 21st day in the spleen and hind paw joint tissue homogenates. The tissues were homogenized for 1 min at 0 °C using Ultra Turax TP 18/10 (Janke and Kunkel) in a phosphate buffer (pH 8.1, 2.6 mM NaH_2_PO_4_, 50 mM Na_2_HPO_4_, 68 mM NaCl, 15 mM EDTA). The biochemical substrates were subsequently diluted to final concentrations of 2.5 mM and 12.6 mM, respectively, in isopropyl alcohol (65%). The biochemical substrates were 44 mM of methionine and 8.7 mM of l-glutamyl-p-nitroanilide. The samples were incubated for an hour at 37 °C before the reaction was stopped by adding 2.3 mL of cold methanol. The Eppendorf centrifuge was used to centrifuge the tubes for 20 min at a centrifugal force of 1957× *g* (rotor radius = 7 cm at 5000 rpm). The absorbance of the supernatant (product p-nitroaniline) was determined at 406 nm using a spectrophotometer Specord 40 (Analytikjena, Jena, Germany). As blanks, solution mixtures with or without a substrate or an acceptor were utilized. Based on an absorbance measurement and a calibration coefficient, the activity was determined.

### 2.8. Isolation of mRNA, Reverse Transcription, and Quantitative Real-Time PC

Total mRNA isolation from rat livers using RNAzol^®^ RT (M.R.C. Inc., Cincinnati, OH, USA) and qRT-PCR (Solis BioDyne, Tartu, Estonia) performed on QuantStudio™ 3 thermocycler (Applied Biosystems, Thermo Fisher Scientific, Foster City, CA, USA) has already been extensively addressed by Chrastina et al. (2022) [41]. Takara Primescript™ RT Reagent Kit was used to reverse transcribe isolated mRNA into cDNA in accordance with the manufacturer’s instructions. With the use of HOT FIREPol EvaGreen qPCR Mix Plus and the primers for CD36 and IL-1β, cDNA was amplified. The endogenous control, β-actin, was used to evaluate the relative mRNA expression using the ΔΔCt calculations.

### 2.9. Statistical Evaluation of the Experimental Results

Arithmetic mean (average value) and ± standard error of the mean (SEM) was applied to express the experimental results. The statistically significant differences between the experimental groups were determined using the GraphPad InStat3 program. ANOVA was used to evaluate any significant differences between the control animals (HC), untreated animals (AA), and the treatment groups of animals (MTX, SF1, SF2, SF1 + MTX, and SF2-MTX). In cases where there were significant variations between the groups, the post hoc test (Tukey–Kramer) was used. Following the post hoc screening, the following degrees of significance were established: not significant (*p* > 0.05); significant (*p* ≤ 0.05); very significant (*p* ≤ 0.01); and highly significant (*p* ≤ 0.001). The legend located beneath each table and graphic provides information about the specific symbol of significance.

## 3. Results

### 3.1. Biometric Parameters

All biometric parameters were measured on days 14 and 21 of the experiment for the monitoring of arthritis development and of the animals’ welfare condition. 

#### 3.1.1. The Change of the Animal Body Weight

Following both experimental days 14 and 21, the parameter change of body weight (ChBW) was considerably lower in the untreated AA animal group compared to the healthy control (HC) group (Figure 1; +++ *p* ≤ 0.001 AA vs. HC). On day 14, animals in the methotrexate (MTX) group and the MTX group in combination with the saffron extract—both doses (SF1 + M and SF2 + M)—weighed substantially more than untreated AA animals (Figure 1; *** *p* ≤ 0.001 vs. AA). On day 14, there was no apparent difference between the MTX group and the SF1 + M and SF2 + M groups (Figure 1). Both doses of saffron monotherapy (SF1 and SF2) significantly increased the weight of animals compared to the untreated AA on day 14 (Figure 1; *** *p* ≤ 0.001 vs. AA for SF1 and * *p* ≤ 0.05 vs. AA for SF2). On day 21, a similar significant increase in ChBW was noticed in the following groups: MTX, SF1 + M, and SF2 + M, compared to the untreated AA group (Figure 1; *** *p* ≤ 0.001 vs. AA). 

#### 3.1.2. The Change in Hind Paw Volume of Experimental Animals

The parameter, change in the hind paw volume (CHPV), was measured on days 14 and 21 (Figure 2). A significant difference between the HC and AA groups was observed on day 14 (Figure 2; +++ *p* ≤ 0.001 AA vs. HC). Methotrexate applied in monotherapy and in combination with saffron in both doses (SF1 + M and SF2 + M) significantly decreased the hind paw volume compared to the untreated AA group on day 14 (Figure 2; *** *p* ≤ 0.001 vs. AA). Furthermore, the monotherapy of a higher dose of saffron (SF2) significantly decreased the hind paw volume in the SF2 group compared to the untreated AA group on the 14th day (Figure 2; *** *p* ≤ 0.001 vs. AA), but the lower dose of saffron (SF1) was without a significant effect. The CHPV was significantly increased in the untreated AA group compared to the HC group on day 21 of the experiment (Figure 2; +++ *p* ≤ 0.001 AA vs. HC). Groups for which the combinations SF1 + M and SF2 + M were applied demonstrated a significant effect on CHPV compared to the untreated AA group on day 21 (Figure 2; *** *p* ≤ 0.001 vs. AA). Moreover, the combination of the higher dose of saffron (SF2) and MTX had a significant effect compared to the MTX monotherapy on day 21 (Figure 2; ### *p* ≤ 0.001 vs. AA-MTX). Both monotherapeutic doses of saffron extract, as well as MTX monotherapy, were without a significant effect on CHPV compared to the untreated AA group (Figure 2). 

#### 3.1.3. The Course of the Arthritic Score

The biometric multi-parameter arthritic score (AS) was significantly increased in the untreated AA group compared to the HC group on 14th and 21st days (Figure 3; +++ *p* ≤ 0.001 AA vs. HC). The monotherapy with administered MTX, SF1, or SF2 significantly decreased AS compared to the untreated AA group on the 14th and 21st days (Figure 3; *** *p* ≤ 0.001 and * *p* ≤ 0.05 vs. AA). The therapy with both doses of saffron in combination with MTX significantly decreased AS compared to untreated AA animals during both experimental days (Figure 3; *** *p* ≤ 0.001 vs. AA). Moreover, the combination therapy of SF1 + M and SF2 + M significantly decreased AS compared to MTX administered in monotherapy on day 21 (Figure 3; # *p* ≤ 0.05 and ### *p* ≤ 0.001 vs. AA-MTX). All treated experimental groups demonstrated a very similar significant pattern on AS compared to the untreated AA group (Figure 3).

### 3.2. The Activity of Gamma-Glutamyl Transferase Activity Measured in Relevant Tissues

#### 3.2.1. Gamma-Glutamyl Transferase Activity in the Joint

The gamma-glutamyl transferase (GGT) activity was evaluated on day 21 of the experiment in the tissue homogenate of the hind paw joint. In the untreated AA group of animals, significantly increased activity of GGT in the joints was observed compared to the HC experimental group (Table 2; ++ *p* ≤ 0.01 AA vs. HC). The activity of GGT was significantly lowered by MTX monotherapy, as well as by both saffron combination therapies (SF1 + M and SF2 + M), compared to the AA group with any given treatment (Table 2; ** *p* ≤ 0.01 vs. AA, * *p* ≤ 0.05 vs. AA). Both monotherapies with SF1 and SF2 significantly decreased GGT activity in the joints compared to the untreated AA group, demonstrating a similar pattern (Table 2; ** *p* ≤ 0.01 vs. AA, * *p* ≤ 0.05 vs. AA). 

#### 3.2.2. Gamma-Glutamyl Transferase Activity in the Spleen 

The gamma-glutamyl transferase (GGT) activity in the spleen was measured on the 21st experimental day. The untreated AA group significantly increased the GGT activity in the spleen when compared to the HC group (Table 3; ++ *p* ≤ 0.01 AA vs. HC). The monotherapy of MTX, as well as a lower dose of saffron (SF1), decreased the GGT activity compared to the untreated AA experimental group but not significantly (Table 3; *p* > 0.05 vs. AA). The same insignificant pattern was observed for monotherapy via a higher dose of saffron and its combination with MTX (Table 3; *p* > 0.05 vs. AA). Only the combination of MTX with a lower dose of saffron was able to decrease the GGT activity significantly (Table 3; *** *p* ≤ 0.001 vs. AA). 

### 3.3. Inflammatory Markers Measured in Plasma

#### 3.3.1. Levels of Interleukin 17A

On day 21, AA significantly increased the plasmatic level of interleukin 17A (IL-17A) compared to the HC group (Figure 4; +++ *p* ≤ 0.001 AA vs. HC). Both monotherapies, MTX and SF2, showed a significantly decreased level of plasmatic IL-17A compared to the untreated AA experimental animals (Figure 4; *** *p* ≤ 0.001 vs. AA). The combination therapy of SF1 + M and SF2 + M significantly lowered the levels of plasmatic IL-17A compared to the untreated AA animals (Figure 4; *** *p* ≤ 0.001 vs. AA). SF1 also lowered the level of IL17A but without significance (Figure 4; *p* > 0.05 vs. AA).

#### 3.3.2. Plasmatic Levels of Interleukin 1β 

Plasmatic levels of interleukin 1β (IL-1β) were evaluated on the 21st experimental day in plasma. The level of plasmatic IL-1β was significantly increased in the untreated AA experimental animals compared to the HC group (Figure 5; +++ *p* ≤ 0.001 AA vs. HC). The monotherapies with MTX, SF1, and SF2, as well as the therapies with the combinations MTX and SF1 and MTX and SF2, significantly decreased the levels of plasmatic IL-1β compared to the untreated AA animals (Figure 5; *** *p* ≤ 0.001 for SF2 + M, ** *p* ≤ 0.01 for SF2 and * *p* ≤ 0.05 for MTX, SF1, and SF1 + M vs. AA). 

#### 3.3.3. Plasmatic Levels of C-Reactive Protein

On day 21 of the experiment, plasma was examined for the presence of C-reactive protein (CRP). When compared to the HC group, the level of CRP on day 21 was considerably higher in the untreated AA group (Table 4; +++ *p* ≤ 0.001 AA vs. HC). Except for the combination of SF2 and MTX, which exhibited a major decrease in the CRP level on day 21 (Table 4; * *p* ≤ 0.05, SF2 + M vs. AA), the other treated groups demonstrated no apparent effect on the level of CRP in plasma compared to the untreated AA group on day 21 (Table 4). 

#### 3.3.4. Plasmatic Levels of Matrix Metalloproteinase-9

The plasmatic levels of matrix metalloproteinase-9 (MMP-9) were measured on the 21st experimental day. The level of MMP-9 was significantly increased in the untreated AA animals compared to the HC (Figure 6; +++ *p* ≤ 0.001 AA vs. HC). Both monotherapies, SF1 and SF2, as well as MTX monotherapy, significantly decreased the plasmatic level of MMP-9 compared to the untreated AA animals (Figure 6; *** *p* ≤ 0.001 for MTX and ** *p* ≤ 0.01 for SF1 and SF2 vs. AA). The combination of SF1 + M and SF2 + M significantly lowered the plasmatic level of MMP-9 compared to the untreated AA experimental group (Figure 6; *** *p* ≤ 0.001 vs. AA).

### 3.4. Expression of Hepatal IL-1β and CD36/FAT Gene mRNA

#### 3.4.1. Levels of Relative mRNA IL-1β Expression

We observed a significant increase in hepatal IL-1β gene expression in untreated AA animals compared to HC animals (Figure 7; +++ *p* ≤ 0.001, HC vs. AA). The saffron extract significantly attenuated IL-1β mRNA expression in both doses (Figure 7; *** *p* ≤ 0.001, SF1 vs. AA; *** *p* ≤ 0.001, SF2 vs. AA). MTX monotherapy decreased the gene expression of IL-1β with a weaker effect than both saffron extracts applied in monotherapies (Figure 7; * *p* ≤ 0.05, MTX vs. AA). Despite the anti-inflammatory impact of both saffron extracts, the combination of MTX and both saffron extracts led to a weaker effect on this parameter, (Figure 7; ** *p* ≤ 0.01, SF2 + M vs, AA; ** *p* ≤ 0.01, SF1 + M vs. AA).

#### 3.4.2. Relative mRNA Expression of CD36/FAT

In the untreated AA group, there was a significant increase in the liver gene expression of the cluster of differentiation 36/fatty acid translocase (CD36/FAT; Figure 8; +++ *p* ≤ 0.001, HC vs. AA) compared to the HC group. The lower (SF1), but not the higher dose of saffron extract monotherapy, significantly decreased the mRNA expression of CD36 (Figure 8; ** *p* ≤ 0.01, SF1 vs. AA). The monotherapy of MTX lowered the mRNA expression of CD36 statistically insignificantly in comparison to the untreated AA group (Figure 8). The lower-dosage saffron extract (SF1) in combination with MTX resulted in the same pattern as the SF1 monotherapy in comparison to the untreated AA group (Figure 8; ** *p* ≤ 0.01, SF1 + M vs. AA). 

## 4. Discussion

Autoimmune diseases such as rheumatoid arthritis (RA) remain challenging topics for scientists worldwide. Epidemiological studies have shown that the prevalence of RA in Europe and North America ranges from 0.5% to 1% [5,6,42,43]. Even though RA does not represent a direct threat to the life of patients, it is associated with many complications which can lead to the failure of vital organs [44]. RA influences many aspects of the daily life of patients; therefore, it has not only physical consequences but also psychic and socioeconomic impacts [45]. From a pathophysiological point of view, RA is an autoimmune and inflammatory disease, which means that the body’s immune system attacks healthy cells by mistake, causing inflammation (painful swelling) in the affected body parts [46,47]. It is characterized mainly by inflammation of the joints that prograde damage to other organs such as the lungs, liver, and spleen [48]. Due to the aspects mentioned above, there is still an unmet medical need for safe and effective RA management. In the last 50 years, the development of RA treatment has gone from synthetic drugs to biological therapy. However, treatment is still mainly focused on symptoms because the pathophysiological origin of RA remains unknown. Current RA therapy consists of the effective treatment and management of the disease and self-management strategies [49]. Disease-modifying antirheumatic drugs (DMARDs), which reduce the disease’s progression and prevent joint deformity, are typically used to treat RA; biological and targeted DMARDs have been classified as second-line therapies [50]. Non-steroidal anti-inflammatory drugs (NSAIDs) and glucocorticoids have been shown to reduce the pain and inflammation related to RA at the beginning of the therapy, together with conventional synthetic DMARDs (csDMARDs, e.g., methotrexate, sulfasalazine, leflunomide, hydroxychloroquine, etc.), which are the first line of treatment [51]. Among the most prescribed csDMARDs, methotrexate is consistently considered to be the gold standard for RA therapy [52]. On the other hand, in recent years, natural products’ anti-inflammatory and immunomodulatory activities, which are able to control autoimmune inflammation such as RA, have been widely investigated [53,54]. In this original article, we describe the effects of both, 1. the best features of MTX antiarthritic activity and 2. the anti-inflammatory properties derived from natural compounds; in our case, *Crocus sativus* L. (saffron, SF) extract. To observe the dose-dependen manner, we tested two doses of SF in a monotherapeutical setting and the same doses of SF in combination with MTX (Table 1).

To investigate the possible antiarthritic and anti-inflammatory properties of saffron extract, we monitored the ability of saffron extract to significantly modify biometric parameters such as the change of body weight, the change of hind paw volume, and the scale the arthritic score (see Section 3.1: Biometric parameters) in the adjuvant arthritis model. The parameter changes in body weight from both monotherapeutic doses on day 14 were significantly modified (Figure 1). Our results also agree with study results obtained on rats with streptomycin-induced diabetes followed by saffron extract therapy. In this case, the authors noted a significant increase in the weight of rats to which saffron extract was administered [55]. Another reason might be that AA also results in increased stress levels, which are also reflected in weight changes. Halataei et al. (2011) observed an increase in weight in a model of stress-induced anorexia mice administered aqueous saffron extract [56]. Improvement in clinical parameters was also observed in clinical trials with saffron administration in patients with RA, suggesting a beneficial effect of saffron on RA in general [57]. On the other hand, monotherapy did not show any significant increase in body weight for both doses of SF on day 21 (Figure 1). To explain this, it should be noted that adjuvant arthritis is a model of inflammatory cachexia, which is more visible over time [58]. We hypothesize that this insignificant effect on weight gain at the end of the experiment might be caused by the ability of saffron to lower plasma triglyceride (TG), reduce the low-density lipoprotein (LDL) cholesterol, and lower cholesterol concentrations in the long term, therefore deepening the cachexia caused by induced arthritis. This hypothesis is supported by Rahmani et al. (2019), who describe the dose–response effect of saffron on the weight and lipid profile reduction described in a systematic review and meta-analysis of randomized clinical trials [59]. Taking into account the severity of biometric manifestations on the AA model, it could be concluded that at the beginning of the SF administration, saffron had a positive effect on weight gain until day 14; after day 14, the weight was decreased by the action of both the SF effect via TG and LDL reduction and the arthritic cachexia itself as mentined above. Methotrexate at therapeutic doses does not affect sarcopenia or cachexia in RA [60,61]. However, in AA, we found that MTX at sub-therapeutic doses was able to increase the body weight of AA animals. This finding has also been described in the literature with respect to MTX combination in AA/RA [35]. Jurčovičová et al. (2009) also found that a sub-therapeutic dose of MTX significantly increased the body weight of experimental animals [62]. In our experiment, we also recorded a significant increase in the body weight of AA animals which were administered with MTX (Figure 1). No significant differences were determined when comparing MTX monotherapy to its combination with saffron extract (both doses) on either day 14 or day 21 (Figure 1). This result indicates that MTX’s impact on ChBW has already peaked. Saffron extract added to MTX was therefore unable to cause a rise in ChBW. Sahebari et al. (2021) reported similar outcomes in RA patients treated with MTX and saffron (100 mg saffron pill/day) [63]. Moreover, it was also noted that saffron plus damask rose petals reduced weight gain, TG, and CRP (all *p* < 0.05) compared to the control group in a study using diabetic male Sprague Dawley rats [64]. It is also known that saffron supplementation can alleviate DM2 by improving the glycaemic status, lipid profile, liver enzymes, and oxidative status [65]; however, the mechanism of the effect of saffron on the modification of the weight in AA and/or RA remains unclear.

Swelling and stiffness of the joints is the first manifestation of RA, which leads to irreversible damage to the connective tissue and even the bone itself. On day 14, we observed a significant reduction in swelling in the hind limb joint for the higher dose of SF (SF2 group; Table 2). Additionally, MTX alone and in combination with SF1 significantly reduced swelling on day 14 (Figure 2). However, on day 21, more interestingly, the combination of MTX and SF2 led to an improvement in this parameter compared to MTX alone (Figure 2). Our findings are consistent with the findings of other authors with respect to combined therapy, where more significant effects of the combination of ibuprofen and crocin have been described [66]. In our case, we may consider this as an antiarthritic effect of SF in combination with MTX due to the doubled significant evidence (Figure 2). Colleagues from the Bulgarian Academy of Sciences also described an improvement in the pathophysiological state of the connective tissue in the OA model, and based on this information, we selected this dose in our experiment on AA [22].

On day 14 of the experiment, the arthritis score was reduced not only in the MTX, SF2, MTX + SF1, and MTX + SF2 groups but also in the SF1 group (Figure 3). Moreover, on day 21, the significant antiarthritic effect of both combinations (SF1 and SF2 with MTX) was more visible due to the doubled significant evidence (Figure 3). Given that one of the parameters evaluated in the arthritic score is the evaluation of the wound or the scabs formed on the tail after AA induction, we assume that the effect in this case could be due to the ability of saffron to improve wound healing. This hypothesis could be explained by increasing the migration of neonatal human dermal fibroblasts or by saffron’s ability to reduce ROS production [67]. Moreover, this effect might be also due to the ability of the saffron extract to promote scratch wound closure of keratinocytes and to enhance VEGF production [68].

GGT is a reliable marker of AA progression [69]. We had already found a significant decrease in GGT activity in joints via monotherapy with SF1 and SF2 (Table 2). The subsequent addition of both saffron doses to MTX showed similar results, i.e., a significant decrease in GGT activity in the joint (Table 2). We assume that by the action of MTX alone, the activity of GGT reached the level of control; thus, the increase in the effectiveness of the combination could no longer be manifested. For the activity of GGT in the spleen, we observed only a significant decrease result with the combination of SF1 with MTX (Table 3). It has been found that trans-sodium crocetinate can lower the TNF-α level in the liver and spleen and the IL-10 level in the spleen in haemorrhagic rats [70]. Immunohistochemical staining and flow cytometry showed reduced macrophage counts in MLNs and the spleen as well as the infiltration of macrophages in colonic tissues [71]. Therefore, the effect of SF (Table 3) might be explained by the reduced chemokines level and the reduced macrophage infiltration in the spleen. Note that a similar significant effect of a lower SF dose (SF1) in combination with MTX was also observed on the level of IL-17A (Figure 4) and MMP-9 (Figure 6) in plasma on day 21, as well as on the relative mRNA expression of CD36 in the liver (Figure 8). Unfortunately, this does not explain the significant effect of the lower dose of SF.

The saffron extract in our experiment on monotherapy at a higher dose and in combination with MTX (Figure 4) significantly reduced IL-17A levels, which is in accordance with the current scientific literature. A pilot study carried out on patients with ulcerative colitis showed improvement in patients given 50 mg of saffron extract twice daily. There was a reduction in important inflammatory markers, including IL-17A [72]. Saffron extract was the subject of a different investigation that looked at how it affected the clinical and immunological characteristics of experimental autoimmune diabetes in C57BL/6 mice. Treatment with saffron decreased the lymphocyte proliferation index in pancreatic cells isolated from diabetic mice. A reduction in IL-17A levels has been described in a murine model of autoimmune diabetes. The saffron extract significantly increased the anti-inflammatory IL-10 and transforming growth factor-β in the pancreatic cell population, and it reduced the production of pro-inflammatory interleukin-17. Furthermore, the production of pro-inflammatory nitric oxide and reactive oxygen species was reduced via saffron extract treatment [73]. Because IL-17A is produced by Th17 cells, it also affects the stimulation of macrophages, fibroblasts, and chondrocytes. In our experiment, there was a reduction in hind limb swelling as well as an improvement in the arthritic score that was driven by saffron, and we assume that one of the mechanisms could be the influence of the IL-17A signalling pathway. At the same time, we were unable to find authors in the current literature who present similar data with respect to IL-17A levels in connection with the AA model. However, the immunoregulatory and anti-inflammatory properties of *Crocus sativus* L. have been widely described for other interleukins by Zeinali et al. (2019) [74].

Inflammatory cell types such as synovial fibroblasts produce MMP-9. MMP-9 activates TNF- and IL-6, two RA-determining factors, leading to inflammation and bone and cartilage disintegration [75,76]. The therapeutic effects of saffron extract have also been found in patients suffering from multiple sclerosis (MS). Using ELISA kits, the serum levels of MMP-9 and TIMP-1, which is MMP-9′s inhibitor, were determined. After 12 months of saffron extract oral therapy, the serum levels of MMP-9 in MS patients decreased considerably (*p* = 0.006), although the changes were not significant before and after 12 months of therapy compared to placebo. No significant change was seen between the time before and after administration of placebo pills, even though TIMP-1 levels significantly rose after one year of therapy with saffron (*p* = 0.0002). According to the study’s findings, taking saffron extract orally for 12 months could significantly lower the level of MMP-9 in the blood and increase the TIMP-1 levels in MS patients [77]. In our experiment, we determined a significant decrease in the level of MMP-9 for all therapeutic groups, i.e., SF1, SF2, SF1 + MTX, and SF2 + MTX, as well as MTX (Figure 6). An experiment performed on rats to which AA was induced and which were subsequently administered with crocin, one of the metabolites contained in the saffron extract, reported a decreased level of MMP-9 [66]. In our experiment, we achieved a similar effect; however, there were minor significant changes between the treated groups. Therefore, we assume that with an increased dose of the extract, we would observe a similar effect as observed by the Hemshekhar et al. (2012) [66].

In a mouse model of RA, Rathore et al. (2015) administered three dosages of crocin (25, 50, and 100 mg/kg) over the course of 47 days. When greater dosages of crocin were provided, they saw a reduction in TNF- and IL-1 levels and an increase in superoxide dismutase (SOD) and glutathione reductase activity [78]. Rats with experimental arthritis received 140 mg/kg of crocin daily for 14 days in an experiment by Hu et al. (2019). When compared to controls, the crocin-treated rats had far less swelling in their hind limbs and had smaller ankle diameters. Additionally, a histological investigation revealed that other organs, such as the spleen and joints, were less inflamed. Moreover, synovial tissues showed a reduction in TNF- and TGF-1 levels [79]. The chondrogenic effects of crocin were shown in a study by Ding et al. (2013), where rabbits received a 0.3 mL intra-articular injection of crocin (5 and 100 μM, respectively) [80]. According to their research, crocin reduced the synthesis of MMP-1, -3, and -13 in chondrocytes and lowered the expression of IL-1β, perhaps via blocking the NF-κB pathway. Their findings demonstrated that crocin may minimize cartilage deterioration in the knees of rabbits with induced OA [80]. In agreement with the study mentioned above by Ding et al. (2013), a higher concentration of saffron extract (SF2) reduced the level of IL-1β (Figure 5) in our model. However, the amount of crocin in the extract itself cannot be directly compared. Our results for plasma IL-1β on day 21 showed that saffron extract monotherapy, as well as in combination with MTX, was as effective as MTX monotherapy in reducing plasma IL-1β (Figure 5), which could be an interesting insight in the treatment of RA patients.

An investigation by Hamidi et al. (2020) intended to discover the impact of supplementary saffron extract therapy on clinical outcomes and metabolic profiles in RA patients. In this randomized, double-blind, placebo-controlled experiment, 66 female participants over the age of 18 were given either a 100 mg/day saffron supplement (intervention group; n = 33) or a placebo (placebo group; n = 33) for a period of 12 weeks. The saffron group’s C-reactive protein levels were lower than they had been at the beginning of the treatment. The levels of IFN-γ, TNF-α, and malondialdehyde were also reduced. According to the authors of the study, supplemental treatment with saffron could improve the clinical manifestations of arthritis in RA patients [57]. Ghaderi et al. (2020), in a meta-analysis, described the effect of saffron on mental health parameters and C-reactive protein [81]. They reported that saffron alone had no significant effect on CRP change, which is in agreement with our results for saffron extract monotherapies (Table 4). In our experiment, the only significant decrease in CRP level was on day 21 in the group receiving combined therapy of a higher dose of saffron extract (SF2) and MTX (Table 4). 

According to our earlier findings, the relative mRNA expression of IL-1β in the liver of untreated AA rats increased in comparison to healthy animals. Moreover, MTX’s weak or missing effect on IL-1β gene expression has been repeatedly reported [82,83]. The saffron extract (SF1 and SF2) significantly attenuated this parameter, even in the monotherapy (Figure 7). Based on the literature, saffron is able to suppress the degradation of the NF-κB inhibitor Iκκ and thereby inhibit the NF-κB pro-inflammatory pathway in addition to JNK in different arthritic models, resulting in a decreased expression of proinflammatory cytokines including IL-1β [66,84,85]. The strong significant effect of the saffron extract on IL-1β mRNA expression in our experiment (Figure 7) might also be mediated by the suppression of the STAT3 activation [86]. Moreover, the activated LXRα in the rat AA model has been shown to upregulate NF-κB signalling [87]. In agreement with Xie et al. (2021) [87], the LXRα-dependent gene CD36 was upregulated in the liver of untreated arthritic rats in our study (Figure 8). Hepatic CD36 is a membrane receptor involved in fatty acid uptake into the liver; it is responsible for hepatic steatosis and associated with systemic inflammation [88,89]. Hence, CD36 is considered to be a crucial aspect of progression to liver steatohepatitis [90]. The increased fatty acid uptake and oxidation estimated in the livers of untreated AA rats indicate enhanced catabolic processes in the liver leading to cachexia and dyslipidaemia, as observed in arthritic animals [91] and RA patients [92]. The saffron extract, either with or without MTX, decreased CD36 gene expression (the lower dose significantly; the higher dose non-significantly), which might result in lower hepatic FA uptake and contribute to lipid homeostasis. The hypolipidemic potential of a picrocrocin-enriched fraction (obtained from an extract of saffron made from the stigmas of *Crocus sativus* L.) was examined in HepG2 cells in a study by Frattaruolo et al. (2023) [93]. The authors of this study offered new perspectives on the metabolic effects of picrocrocin, which appear to be mediated by a mechanism distinct from that of statins [93]. However, the precise mechanism of saffron’s action on the lipid metabolism in arthritic livers needs to be elucidated.

## 5. Conclusions

Since *Crocus sativus* L. is recognized mainly as a neuromodulator of psychosomatic behavior, especially in terms of saffron’s significant effect on the severity of depression [94], the antirheumatic and anti-inflammatory properties have barely been investigated. In this experimental article, we have analyzed the biometric, plasmatic, and relative mRNA gene expression parameters in the liver to evaluate saffron’s effects on the adjuvant arthritis model via two doses (as monotherapy) and in combination with methotrexate (MTX). Summarizing all experimental data obtained, we may assume that saffron extract can reduce the joint’s swelling, which is supported by the significant improvement in the arthritic score. Moreover, this antiarthritic result was also significant when compared to MTX in monotherapy. Furthermore, saffron’s anti-inflammatory effect was shown on plasmatic IL-17A, IL-1β, MMP-9, and CRP, with similar statistically significant patterns. On the organ level, relative CD36 and IL-1β mRNA expression in the liver demonstrated the ability of the saffron extract to influence lipid metabolism and inflammation. Interestingly, the relative expression of CD36 was more profound at lower doses of saffron. On the other hand, the dose effect of saffron on relative IL-1β mRNA liver expression has not been shown. However, this parameter showed similarities with the IL-1β concentration in plasma, indicating the same mechanism of action. For the novelty of this particular experiment, please consult Figure 9.

As the proposed novel effect (Figure 9), the saffron’s impact should be further investigated as a part of the potential treatment strategy for RA as well. We plan to conduct additional in vitro research on both, the cellular and the in vivo level to determine the precise mode of action of saffron extract. Moreover, we would like to elucidate the dose effect via pharmacokinetic analysis and study the mechanism of action on the level of the main secondary metabolites which are presented in our saffron extract.

## Figures and Tables

**Figure 1 nutrients-15-04108-f001:**
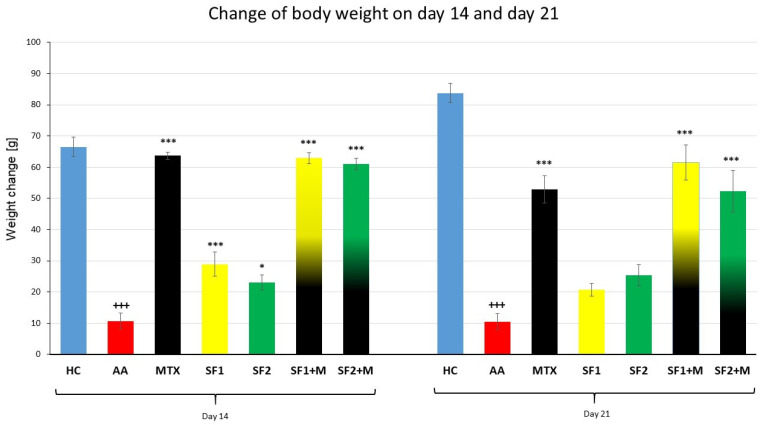
The change of animal body weight was measured on days 14 and 21 of the experiment. The experimental animals were divided as follows: HC—healthy control; AA—adjuvant arthritis; MTX—methotrexate in the dose of 0.3 mg/kg twice weekly; SF1—saffron dosage 25 mg/kg a day; SF1 + M—saffron dosage 25 mg/kg a day in combination with MTX 0.3 mg/kg twice weekly; SF2—saffron dosage 50 mg/kg a day; SF2 + M—saffron dosage 50 mg/kg a day in a combination with MTX 0.3 mg/kg twice weekly. Experimental results were presented as a mean ± S.E.M., n = 8–9 animals per group. ANOVA was performed to evaluate statistical significance for independent variables. The symbols representing the significant change were the following: +++ *p* ≤ 0.001 AA vs. HC; *** *p* ≤ 0.001; and * *p* ≤ 0.05 vs. AA.

**Figure 2 nutrients-15-04108-f002:**
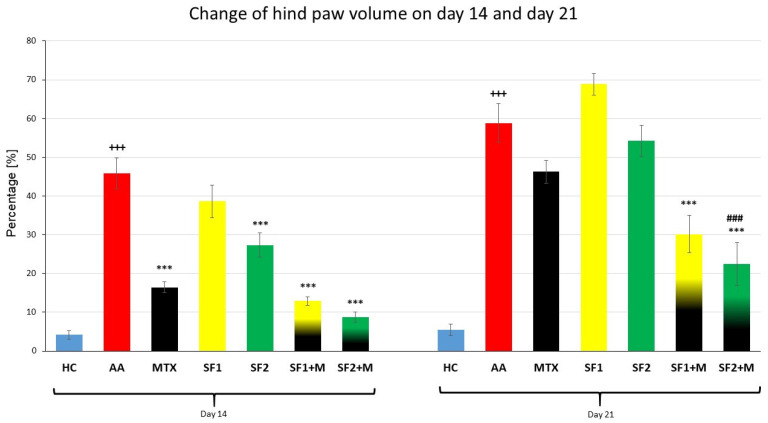
Change of hind paw volume was measured on days 14 and 21 of the experiment. The experimental animals were divided as follows: HC—healthy control; AA—adjuvant arthritis; MTX—methotrexate in the dose of 0.3 mg/kg twice weekly; SF1—saffron dosage 25 mg/kg a day; SF1 + M—saffron dosage 25 mg/kg a day in combination with MTX 0.3 mg/kg twice weekly; SF2—saffron dosage 50 mg/kg a day; SF2 + M—saffron dosage 50 mg/kg a day in a combination with MTX 0.3 mg/kg twice weekly. Experimental results were presented as a mean ± S.E.M., n = 7–9 animals per group. ANOVA was performed to evaluate statistical significance for independent variables. The symbols representing the significant change were the following: +++ *p* ≤ 0.001 AA vs. HC; *** *p* ≤ 0.001 vs. AA; ### *p* ≤ 0.001 vs. AA-MTX.

**Figure 3 nutrients-15-04108-f003:**
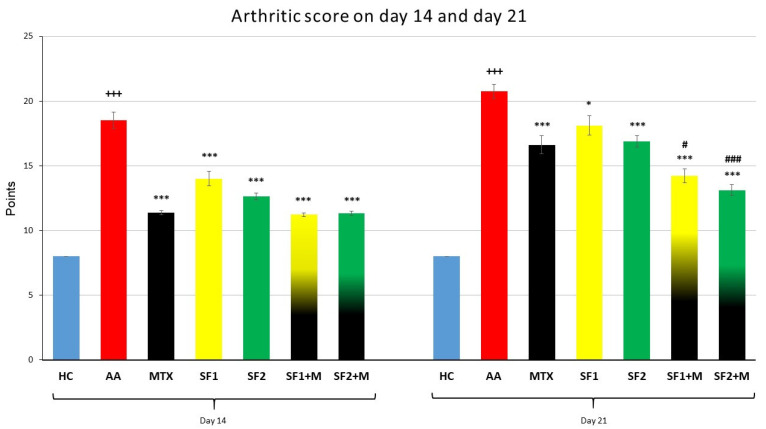
Arthritic score was measured on days 14 and 21 of the experiment. The experimental animals were divided as follows: HC—healthy control; AA—adjuvant arthritis; MTX—methotrexate in the dose of 0.3 mg/kg twice weekly; SF1—saffron dosage 25 mg/kg a day; SF1 + M—saffron dosage 25 mg/kg a day in combination with MTX 0.3 mg/kg twice weekly; SF2—saffron dosage 50 mg/kg a day; SF2 + M—saffron dosage 50 mg/kg a day in a combination with MTX 0.3 mg/kg twice weekly. Experimental results were presented as a mean ± S.E.M., n = 8–9 animals per group. ANOVA was performed to evaluate statistical significance for independent variables. The symbols representing the significant change were the following: +++ *p* ≤ 0.001 AA vs. HC; *** *p* ≤ 0.001 and * *p* ≤ 0.05 vs. AA; # *p* ≤ 0.05 and ### *p* ≤ 0.001 vs. AA-MTX.

**Figure 4 nutrients-15-04108-f004:**
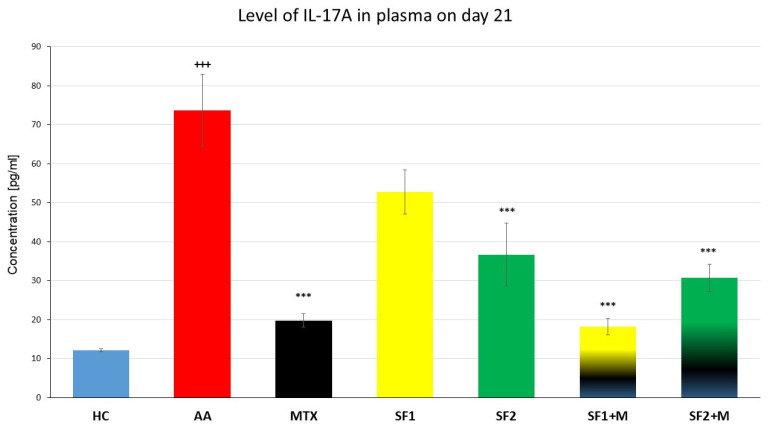
Levels of IL-17A the day 21 of the experiment. The experimental animals were divided as follows: HC—healthy control; AA—adjuvant arthritis; MTX—methotrexate in the dose of 0.3 mg/kg twice weekly; SF1—saffron dosage 25 mg/kg a day; SF1 + M—saffron dosage 25 mg/kg a day in combination with MTX 0.3 mg/kg twice weekly; SF2—saffron dosage 50 mg/kg a day; SF2 + M—saffron dosage 50 mg/kg a day in a combination with MTX 0.3 mg/kg twice weekly. Experimental results were presented as a mean ± S.E.M., n = 7 samples per group. ANOVA was performed to evaluate statistical significance for independent variables. The symbols representing the significant change were the following: +++ *p* ≤ 0.001 AA vs. HC; *** *p* ≤ 0.001 vs. AA.

**Figure 5 nutrients-15-04108-f005:**
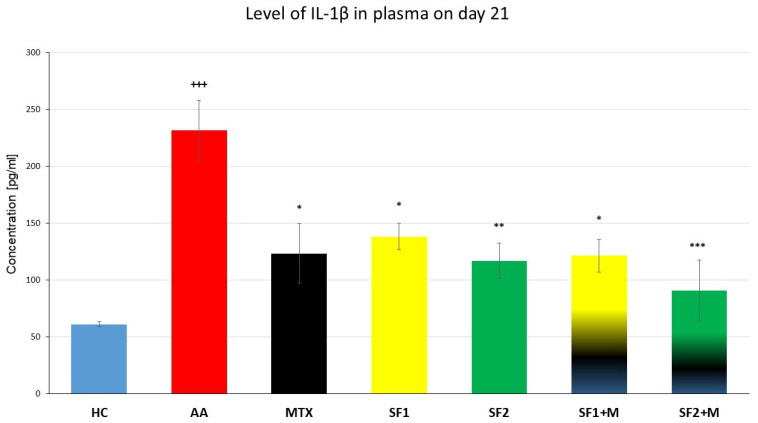
Levels of IL-1β were measured on the day 21 of the experiment. The experimental animals were divided as follows: HC—healthy control; AA—adjuvant arthritis; MTX—methotrexate in the dose of 0.3 mg/kg twice weekly; SF1—saffron dosage 25 mg/kg a day; SF1 + M—saffron dosage 25 mg/kg a day in combination with MTX 0.3 mg/kg twice weekly; SF2—saffron dosage 50 mg/kg a day; SF2 + M—saffron dosage 50 mg/kg a day in a combination with MTX 0.3 mg/kg twice weekly. Experimental results were presented as a mean ± S.E.M., n = 5–7 samples per group. ANOVA was performed to evaluate statistical significance for independent variables. The symbols representing the significant change were the following: +++ *p* ≤ 0.001 AA vs. HC; *** *p* ≤ 0.001, ** *p* ≤ 0.01 and * *p* ≤ 0.05 vs. AA.

**Figure 6 nutrients-15-04108-f006:**
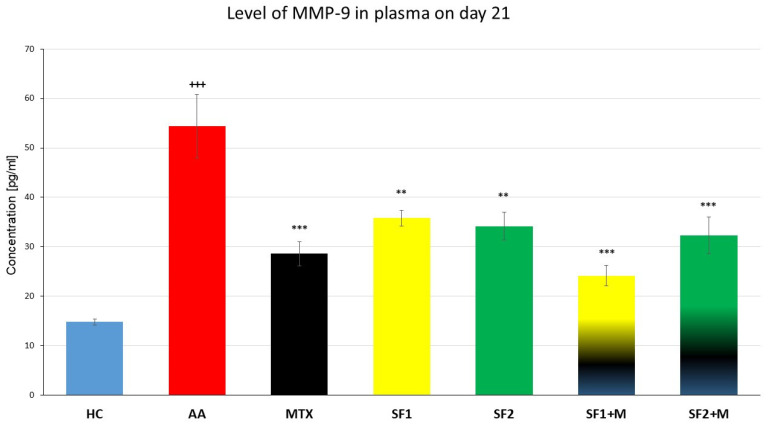
Levels of MMP-9 were measured on the day 21 of the experiment. The experimental animals were divided as follows: HC—healthy control; AA—adjuvant arthritis; MTX—methotrexate in the dose of 0.3 mg/kg twice weekly; SF1—saffron dosage 25 mg/kg a day; SF1 + M—saffron dosage 25 mg/kg a day in combination with MTX 0.3 mg/kg twice weekly; SF2—saffron dosage 50 mg/kg a day; SF2 + M—saffron dosage 50 mg/kg a day in a combination with MTX 0.3 mg/kg twice weekly. Experimental results were presented as a mean ± S.E.M., n = 6–7 samples per group. ANOVA was performed to evaluate statistical significance for independent variables. The symbols representing the significant change were the following: +++ *p* ≤ 0.001 AA vs. HC; *** *p* ≤ 0.001 and ** *p* ≤ 0.01 vs. AA.

**Figure 7 nutrients-15-04108-f007:**
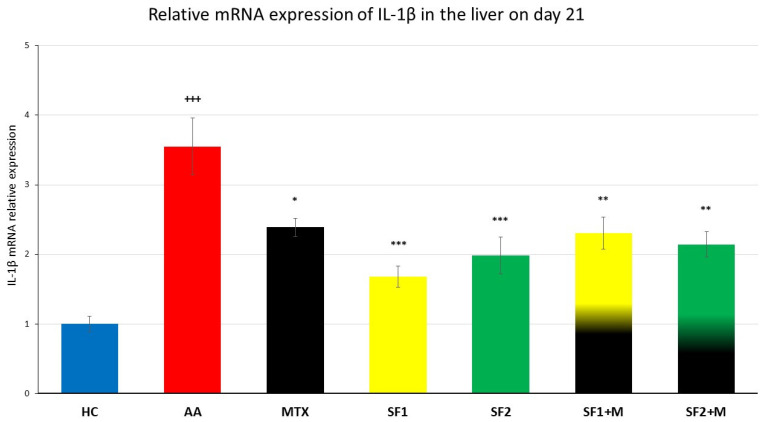
The gene expression of IL-1β in the liver was measured on day 21 of the experiment. The experimental animals were divided as follows: HC—healthy control; AA—adjuvant arthritis; MTX—methotrexate in the dose of 0.3 mg/kg twice weekly; SF1—saffron dosage 25 mg/kg a day; SF1 + M—saffron dosage 25 mg/kg a day in combination with MTX 0.3 mg/kg twice weekly; SF2—saffron dosage 50 mg/kg a day; SF2 + M—saffron dosage 50 mg/kg a day in a combination with MTX 0.3 mg/kg twice weekly. Experimental results were presented as a mean ± S.E.M., n = 6–9 samples per group. ANOVA was performed to evaluate statistical significance for independent variables. The symbols representing the significant change were the following: +++ *p* ≤ 0.001 AA vs. HC; *** *p* ≤ 0.001, ** *p* ≤ 0.01 and * *p* ≤ 0.05 vs. AA.

**Figure 8 nutrients-15-04108-f008:**
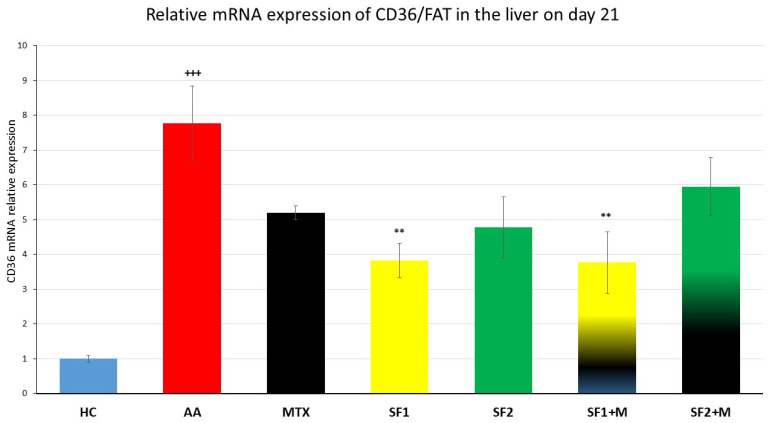
The hepatal gene expression of the cluster of differentiation 36/fatty acid translocase (CD36/FAT) was assessed on day 21 of the experiment. The experimental animals were divided as follows: HC—healthy control; AA—adjuvant arthritis; MTX—methotrexate in the dose of 0.3 mg/kg twice weekly; SF1—saffron dosage 25 mg/kg a day; SF1 + M—saffron dosage 25 mg/kg a day in combination with MTX 0.3 mg/kg twice weekly; SF2—saffron dosage 50 mg/kg a day; SF2 + M—saffron dosage 50 mg/kg a day in a combination with MTX 0.3 mg/kg twice weekly. Experimental results were presented as a mean ± S.E.M., n = 5–7 samples per group. ANOVA was performed to evaluate statistical significance for independent variables. The symbols representing the significant change were the following: +++ *p* ≤ 0.001 AA vs. HC; ** *p* ≤ 0.01 vs. AA.

**Figure 9 nutrients-15-04108-f009:**
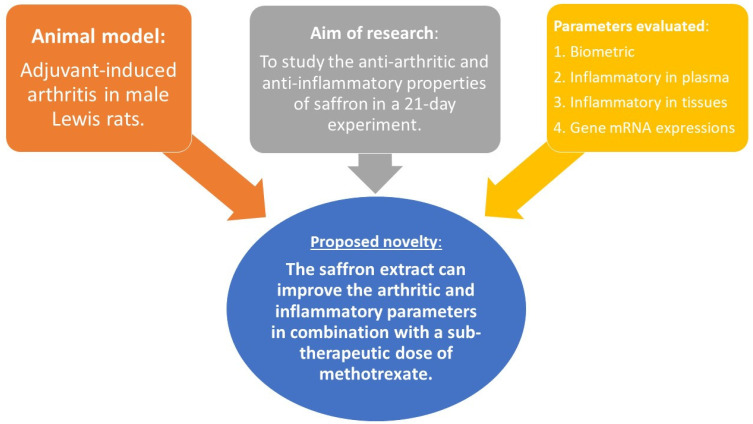
Schematic presentation of the proposed novelty of our in vivo experiment.

**Table 1 nutrients-15-04108-t001:** Experimental design according to the groups.

Name of the Group	Treatment	Dose
Healthy Controls (HC)	Vehiculum	0.5 mL *
Untreated Adjuvant Arthritis (AA)	Vehiculum	0.5 mL *
AA + MTX	Methotrexate (MTX)	0.3 mg/kg **
AA + SF1	Saffron extract (SF1)	25 mg/kg *
AA + SF2	Saffron extract (SF2)	50 mg/kg *
AA + SF1 + MTX	Saffron extract + MTX (SF1 + M)	25 mg/kg * + 0.3 mg/kg **
AA + SF2 + MTX	Saffron extract + MTX (SF2 + M)	50 mg/kg * + 0.3 mg/kg **

* Doses were administered *per os* daily; ** MTX doses were administered *per os* twice a week.

**Table 2 nutrients-15-04108-t002:** The activity of gamma-glutamyl transferase (GGT) in the joint on day 21.

Group	Arithmetic Mean ^1^	SEM ^2^
	[nmol p-nitroaniline/min/g of Tissue]	
HC	4.34	±0.87
AA	10.04 ++	±2.19
MTX	3.69 **	±1.15
SF1	4.33 **	±0.48
SF2	4.67 *	±0.21
SF1 + M	4.77 *	±0.49
SF2 + M	3.66 **	±0.78

The gamma-glutamyl transferase activity was measured on day 21 of the experiment. The experimental animals were divided into following groups: HC—healthy control; AA—adjuvant arthritis; MTX—methotrexate 0.3 mg/kg twice a week; SF1—saffron dosage 25 mg/kg daily; SF1 + M—saffron dosage 25 mg/kg daily in combination with MTX 0.3 mg/kg twice a week; SF2—saffron dosage 50 mg/kg daily; SF2 + M—saffron dosage 50 mg/kg daily in combination with MTX 0.3 mg/kg twice a week. ANOVA was performed to evaluate statistical significance for independent variables. The symbols representing the significant change were the following: ++ *p* ≤ 0.01 AA vs. HC; ** *p* ≤ 0.01 and * *p* ≤ 0.05 vs. AA; ^1^ n = 7; ^2^ standard error of the mean (SEM).

**Table 3 nutrients-15-04108-t003:** The gamma-glutamyl transferase (GGT) activity in the spleen on the day 21.

Group	Arithmetic Mean ^1^	SEM ^2^
	[nmol p-nitroaniline/min/g of Tissue]	
HC	8.93	±1.49
AA	21.77 ++	±2.11
MTX	15.51	±2.61
SF1	17.27	±2.17
SF2	25.06	±2.86
SF1 + M	6.22 ***	±1.37
SF2 + M	19.41	±1.36

The gamma-glutamyl transferase activity was measured on the 21st experimental day. The experimental animals were divided into following groups: HC—healthy control; AA—adjuvant arthritis; MTX—methotrexate 0.3 mg/kg twice a week; SF1—saffron dosage 25 mg/kg daily; SF1 + M—saffron dosage 25 mg/kg daily in combination with MTX 0.3 mg/kg twice a week, SF2-saffron dosage 50 mg/kg daily; SF2 + M—saffron dosage 50 mg/kg daily in combination with MTX 0.3 mg/kg twice a week. ANOVA was performed to evaluate statistical significance for independent variables. The symbols representing the significant change were the following: ++ *p* ≤ 0.01 AA vs. HC and *** *p* ≤ 0.001 vs. AA; ^1^ n = 7; ^2^ standard error of the mean (SEM).

**Table 4 nutrients-15-04108-t004:** Levels of C-reactive protein in plasma.

Group	Arithmetic Mean ^1^	SEM ^2^
	[µg/mL]	
HC	699.77	±180.38
AA	7861.13 +++	±1613.02
MTX	6235.06	±1164.65
SF1	6821.64	±1270.03
SF2	7765.72	±1305.21
SF1 + M	5789.09	±1158.77
SF2 + M	2863.73 *	±566.58

Plasmatic C-reactive protein was measured on the 21st experimental day. The experimental animals were divided as follows: HC—healthy control; AA—adjuvant arthritis; MTX—methotrexate 0.3 mg/kg twice a week; SF1—saffron dosage 25 mg/kg daily; SF1 + M—saffron dosage 25 mg/kg daily in combination with MTX 0.3 mg/kg twice a week; SF2—saffron dosage 50 mg/kg daily; SF2 + M—saffron dosage 50 mg/kg daily in combination with MTX 0.3 mg/kg twice a week. ANOVA was performed to evaluate statistical significance for independent variables. The symbols representing the significant change were the following: +++ *p* ≤ 0.001 AA vs. HC; * *p* ≤ 0.05 vs. AA; ^1^ n = 5–6; ^2^ standard error of the mean (SEM).

## Data Availability

https://figshare.com/articles/dataset/Saffron_extract/23796012. The dataset was posted on the 9 August 2023 and authored by Martin Chrastina.

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
