# Peer review of "Crocus sativus L. Extract (Saffron) Effectively Reduces Arthritic and Inflammatory Parameters in Monotherapy and in Combination with Methotrexate in Adjuvant Arthritis"

_nutrients, 2023, doi:10.3390/nu15194108_

Round 1

Reviewer 1 Report

the authors describe the potential benefits of saffron for the treatment of rheumatoid arthritis. The manuscript is highly interesting but there are several points that should be taken into account.

1. on line 30 during both experimental days....it should be change for something like:  during the experimental period analysed

2. on line 42 where it says ...in mono therapy is more evident.... it should say , whose effect was made stronger in combination with MTX.

In line 54, where it says 700 million people it should say 70 million (it Is 0.5-1% of the world population which is about 8.000.000.000. 700 million is about 10%)

in line 58 where it says , it generates a promoter cascade of local inflammation, please explain the concept the cascade of local inflammation is clear, however the "promoter" is not, it refers to a gene?, please clarify the text

In line 66 were it says disability it should say disabilities

In line 71 were it says medications, consider using drugs

in line 77 were it says injures consider using perturbs (the concept of injuring the homeostasis is a bit complicated)

in line 84 were it says since immemorial change for since time immemorial....

in line 88 were It says by controlling change it for to control....

in line 109 where it days crocetin, picrocrocin change for croceting and picrocrocin...

in line 121 change and for or

in line 152 change in for at

on table 1, the last column has problems with the spacings of the minus sign and the 1 (-1) and it is not clear why on the rows where there it MTX is 0.3 mg/kg-2 were it should most likely be 0.3 mg/kg-1 and then add the mark (2) for the 2 MTX doses were administered per os twice a week. 

in line 222 were it says investigator of common it should say investigators of the common

in line 227 change the for a

in line 234 erase the

in line 242 change and for as

in line 250 please give the g force instead of the rpm since the latter depends on the rotor used

On figure 1: the change of body weight was measured daily as stated on lines 156-157. Thus, show the curves of the daily change of weight for each group, instead of only days 14 and 21, the data will be more complete and allow to see if there are kinetic differences between the monotherapy and the combinations, etc. This is quite important for the interpretation of the data.

Similarly, if for hind paw volume there are additional measurements, please add them to the figure on a line graph 

For the measurements of IL-1ß and IL-17, it is not clear to the referee from the text the reasons for the measuring of these cytokines at such long times. The secretion of these cytokines should show changes on the first 72 hours most likely, and the data obtained at these times (both protein and mRNA) might be easier to interpret that the data shown for 21 days latter.

in the discussion (line 544 and following ones) can you discard whether the increase in weight is due to water retention?. On the other side, it is important to know if the animals from the different groups of treatment eat the same amount of food, in particular since a rat after 24 hours fasting has lost 10% of its total body weight

The changes proposed for English are described in the comments to the authors.

It would improve the easiness of reading if the authors make an effort correcting  some of the expressions used on the manuscript.
The manuscript is highly interesting but there are several points that should be taken into account by the authors before being able to accept the manuscript for publication

Author Response

Author's Reply to the Review Report (Reviewer 1)

Comments and Suggestions for Authors

the authors describe the potential benefits of saffron for the treatment of rheumatoid arthritis. The manuscript is highly interesting but there are several points that should be taken into account.

  1. on line 30during both experimental days....it should be change for something like:  during the experimental period analysed

Authors´ reply: We provided editorial amendments to the above-mentioned, thank you.

  1. on line 42where it says ...in mono therapy is more evident.... it should say , whose effect was made stronger in combination with MTX.

Authors´ reply: Thank you for this comment, we provided editorial amendments to the above-mentioned, thank you.

In line 54, where it says 700 million people it should say 70 million (it Is 0.5-1% of the world population which is about 8.000.000.000. 700 million is about 10%).

Authors´ reply: Thank you for this comment, we provided editorial amendments to the above-mentioned, thank you.

in line 58 where it says , it generates a promoter cascade of local inflammation, please explain the concept the cascade of local inflammation is clear, however the "promoter" is not, it refers to a gene?, please clarify the text

Authors´ reply: Thank you for this comment, we provided editorial amendments to the above-mentioned. The “promoter” was wrongly selected word in that context, therefore we rephrased the sentence to be more precise and clear to the reader, thank you.

In line 66 were it says disability it should say disabilities

Authors´ reply: Thank you for this comment, we provided editorial amendments to the above-mentioned, thank you.

In line 71 were it says medications, consider using drugs

Authors´ reply: Thank you for this comment, we provided editorial amendments to the above-mentioned, thank you.

in line 77 were it says injures consider using perturbs (the concept of injuring the homeostasis is a bit complicated)

Authors´ reply: Thank you for this comment, we provided editorial amendments to the above-mentioned, thank you.

in line 84 were it says since immemorial change for since time immemorial....

Authors´ reply: Thank you for this comment. We have provided editorial amendments to the above-mentioned, thank you.

in line 88 were It says by controlling change it for to control....

Authors´ reply: Thank you for this comment, we provided editorial amendments to the above-mentioned, thank you.

in line 109 where it days crocetin, picrocrocin change for croceting and picrocrocin...

Authors´ reply: Thank you for this comment, we provided editorial amendments to the above-mentioned, thank you.

in line 121 change and for or

in line 152 change in for at

on table 1, the last column has problems with the spacings of the minus sign and the 1 (-1) and it is not clear why on the rows where there it MTX is 0.3 mg/kg-2 were it should most likely be 0.3 mg/kg-1 and then add the mark (2) for the 2 MTX doses were administered per os twice a week. 

in line 222 were it says investigator of common it should say investigators of the common

in line 227 change the for a

in line 234 erase the

in line 242 change and for as

in line 250 please give the g force instead of the rpm since the latter depends on the rotor used

Authors´ reply: Thank you for all these above comments, we have provided necessary corrections and amendments.

On figure 1: the change of body weight was measured daily as stated on lines 156-157. Thus, show the curves of the daily change of weight for each group, instead of only days 14 and 21, the data will be more complete and allow to see if there are kinetic differences between the monotherapy and the combinations, etc. This is quite important for the interpretation of the data.

Authors´ reply: Thank you for this comment. We have measured the body weight daily in order to dose the treated animals exactly. However, these daily measurements were not recorded. We only visually check the body weight to know, if the weight is within the pre-specified margins of the volume we dose the particular animal. For example, if the particular animal weight was from 150g to 199g, we applied 0.20 ml (as a volume with the exact concentration of SF or MTX), if the weight was from 200g to 249 g, we applied 0.25 ml of the extract, etc.

See the below scheme:

Animal’s b.w. (g) ---> volume of the extract applied (ml)

150 g – 199 g --------> 0.20 ml

200 g – 249 g -------> 0.25 ml

250 g – 299 g -------> 0.30 ml

300 g – 349 g -------> 0.35 ml

We recorded only the dedicated days, i.e. day 0 (to see the baseline of the weight) and then the dedicated experimental days, thus day 14 and day 21, in order to calculate the change from the baseline. However, it is a good idea to record all days to see a fine-tuning of the kinetics of the weight change in time. We hope that we have elucidated your concerns. To change the graphs (including day 0), we need permission from the Nutrients editorial board and inform the other two peer-reviewers to be transparent. Consequently, we must add the baseline of day 0 to the submitted raw data, where the editorial board´s permission is also necessary.

Similarly, if for hind paw volume there are additional measurements, please add them to the figure on a line graph 

Authors´ reply: Thank you for this comment. From the statistical point of view, we compare the groups within the particular experimental day, i.e. on day 14 - there we compare the untreated arthritic group with the healthy control group to evaluate the successfully induced arthritis in male Lewis rats. If the arthritis was successfully induced in the untreated group, we may conclude that we may observe statistical differences in the treatment effect. Moreover, the methotrexate group show us the baseline for the standard treatment effect. Thus, if the monotherapy or the combination therapy is significant compared to the methotrexate group alone, the treatment effect is considered superb. The same applies to day 21, which is more closely related to the human RA condition. After more than a 10-year of experience with the adjuvant arthritis model, we consider the current graphical expression of the result as the most relevant. Therefore, in our humble opinion, and considering the statistical rationale and other aspects mentioned above, we wish to have a status quo for the column expression of the biometric parameters.

For the measurements of IL-1ß and IL-17, it is not clear to the referee from the text the reasons for the measuring of these cytokines at such long times. The secretion of these cytokines should show changes on the first 72 hours most likely, and the data obtained at these times (both protein and mRNA) might be easier to interpret that the data shown for 21 days latter.

Authors´ reply: We acknowledge your concerns and the rationale for measuring the cytokines as soon as possible, preferably within the first 72 hours. The cytokines were, however, meant to be as secondary endpoints (parameters) to be observed along with the measurements of gene mRNA expressions of IL1β in the liver. Our primary endpoints (parameters) measured were the biometric parameters, which are the most crucial in human RA, i.e. the hind paw volume in our AA model might be extrapolated to the swollen joint(s) in human RA. Next, the arthritic score (as a multiparametric endpoint) might be extrapolated to the ACR 20%, which is considered as an established diagnostic and prognostic parameter in RA as well as for the evaluation of the treatment effectiveness. The early observation of pro-inflammatory cytokines might be helpful, but in fact, it does not mimic precisely the clinical praxis of the course of the human RA. The observation of pro-inflammatory cytokines, if successful in very early stages (within 72 hours) of the AA (consequently in RA), does not make experimental sense to observe it in the later stages if we take into account the nature of the arthritis. In our case, the 21 day in AA is considered as a later stage and the therapeutic potential of the substances tested exert its therapeutic maximum between day 14 and 21. This has been observed in many AA experiments in the past (see also references: 1.-10.). RA is a lifelong illness, where the stabilisation (or inhibition/neutralisation) of pro-inflammatory cytokines release is also the goal in clinical praxis (e.g. DMARDs, ustekinumab, tocilizumab, etc.) and might be useful during the time. Therefore, observing the pro-inflammatory cytokine releases in time makes more sense in our humble opinion, i.e., the cytokine release stage/condition correlation with the experimental treatment's effectiveness. We hope that this explanation of the association of experimental treatment design on the AA model with clinical praxis might resolve your concerns and missing cytokine result from the 72 h. after the AA induction. However, it is an excellent scientific point made by you. We will discuss the rationale for the further in vivo experiments. In this regard, the team must consider aspects such as the 3Rs, the bridge with clinical praxis and other experimental aspects.

Related references:

  1. Drafi, F.; Bauerova, K.; Chrastina, M.; Taghdisiesfejír, M.; Rocha, J.; Direito, R.; Figueira, M.E.; Sepodes, B.; Ponist, S. Rhodiola rosea L. Extract, a Known Adaptogen, Evaluated in Experimental Arthritis. Mol 2023, 28, 5053
  2. Pružinská, K.; Slovák, L.; Dráfi, F.; Poništ, S.; Juránek, I.; Chrastina, M.; Švík, K.; Strojný, L.; Ambro, Ľ.; Bauerová, K. Enhanced Anti-Inflammatory Effect of the Combination of Lactiplantibacillus plantarum LS/07 with Methotrexate Compared to Their Monotherapies Studied in Experimental Arthritis. Molecules 2022, 28, 297.
  3. Ponist, S.; Zloh, M.; Bauerova, K. Impact of oxidative stress on inflammation in rheumatoid and adjuvant arthritis: damage to lipids, proteins, and enzymatic antioxidant defense in plasma and different tissues. In Animal models in medicine and biology; IntechOpen: 2020.
  4. Bauerova, K.; Ponist, S.; Kuncirova, V.; Mihalova, D.; Paulovicova, E.; Volpi, N. Chondroitin sulfate effect on induced arthritis in rats. Osteoarthritis and Cartilage 2011, 19, 1373-1379.
  5. Bauerova, K.; Ponist, S.; Kuncirova, V.; Drafi, F.; Mihalova, D.; Paulovicova, E.; Volpi, N. Effect of nonanimal high- and low-molecular-mass chondroitin sulfates produced by a biotechnological process in an animal model of polyarthritis. Pharmacology. 2014, 94, 109-14. doi: 10.1159/000366285.
  6. Tsiklauri, L.; Švík, K.; Chrastina, M.; Poništ, S.; Dráfi, F.; Slovák, L.; Alania, M.; Kemertelidze, E.; Bauerova, K. Bioflavonoid robinin from Astragalus falcatus Lam. mildly improves the effect of methotrexate in rats with adjuvant arthritis. Nutrients 2021, 13, 1268
  7. Chrastina, M.; Poništ, S.; Tóth, J.; Czigle, S.; Pašková, Ľ.; Vyletelová, V.; Švík, K.; Bauerová, K. Combination Therapy of Car-nosic Acid and Methotrexate Effectively Suppressed the Inflammatory Markers and Oxidative Stress in Experimental Arthri-tis. Molecules 2022, 27, 7115. https://doi.org/10.3390/molecules27207115.
  8. Jurcovicova, J.; Svik, K.; Scsukova, S.; Bauerova, K.; Rovensky, J.; Stancikova, M. Methotrexate treatment ameliorated testicu-lar suppression and anorexia related leptin reduction in rats with adjuvant arthritis. Rheumatol. Int. 2009, 29, 1187-1191.
  9. Bauerová, K.; Ponist, S.; Ondrejickova, O.g.; Komendová, D.; Mihalová, D. Association between tissue gam-ma-glutamyl-transferase and clinical markers of adjuvant arthritis in Lewis rats. Neuro Endocrinol Lett 2006, 27, 172-175.
  10. Pašková, Ľ.; Kuncírová, V.; Poništ, S.; Mihálová, D.; Nosáľ, R.; Harmatha, J.; Hrádková, I.; ÄŒavojský, T.; Bilka, F.; Šišková, K.; Paulíková, I.; Bezáková, L.; Bauerová, K. Effect of N-Feruloylserotonin and Methotrexate on Severity of Experimental Arthri-tis and on Messenger RNA Expression of Key Proinflammatory Markers in Liver. J. Immunol. Res. 2016, 2016, 7509653.

in the discussion (line 544 and following ones) can you discard whether the increase in weight is due to water retention?. On the other side, it is important to know if the animals from the different groups of treatment eat the same amount of food, in particular since a rat after 24 hours fasting has lost 10% of its total body weight

Authors´ reply: We did a research regarding the potential water homeostasis, while saffron extract application during experimental in vivo condition. In this regard, the only one experimental publication mentions the opposite, i.e. the diuretic effect described by Shariatifar et al., (2014). The authors applied three doses of saffron (1.), while even the lowest dose (60mg/kg of b.w.) already exerted a significant diuretic effect compared to hydrochlorothiazide (*p<0.05). These results are considered as unique but not comprehensive (1.). In our opinion, if any effect observed regarding water homeostasis while saffron extract application, the mechanism of action is still not known. We also do not consider that saffron extract could have an effect on water retention. However, it needs to be studied and confirmed in clinical setting as well. In our humble opinion, the mechanism of weight gain might be due to the neuronal pathways, which are already known. Saffron in general exerts anti-depressive effects (2.) and this change of a mental attitude might be connected with the weight gain. We are also the opinion that the weight gain is due to the anti-inflammatory and anti-arthritic effect of saffron. The weight loose in untreated AA animals is mainly due to the rheumatoid cachexia (3.), disability and inflammation caused by AA (4.), which is visible also in our results.

  1. Shariatifar N, Shoeibi S, Sani MJ, Jamshidi AH, Zarei A, Mehdizade A, Dadgarnejad M. Study on diuretic activity of saffron (stigma of Crocus sativus L.) Aqueous extract in rat. J Adv Pharm Technol Res. 2014 Jan;5(1):17-20. doi: 10.4103/2231-4040.126982.
  2. Tóth, B., Hegyi, P., Lantos, T., Szakács, Z., Kerémi, B., Varga, G., Tenk, J., Pétervári, E., Balaskó, M., Rumbus, Z., Rakonczay, Z., Bálint, E.R., Kiss, T., Csupor, D. The Efficacy of Saffron in the Treatment of Mild to Moderate Depression: A Meta-analysis. Planta Med. 2019, 85, 24-31. doi: 10.1055/a-0660-9565
  3. Walsmith J, Roubenoff R. Cachexia in rheumatoid arthritis. Int J Cardiol. 2002 Sep;85(1):89-99.
  4. Ollewagen T, Powrie YSL, Myburgh KH, Smith C. Unresolved intramuscular inflammation, not diminished skeletal muscle regenerative capacity, is at the root of rheumatoid cachexia: insights from a rat CIA model. Physiol Rep. 2021 Nov;9(22):e15119.

Reviewer 2 Report

In the present paper, Chrastina et report a biological characterization of the effect of saffron extract on a model of rheumatoid arthritis. Authors performed lots of assays suggesting interesting properties of such a natural compound on inflammation and arthritis. The rational behind the project design is convincing and experimental data were reasonably  justified by the authors. However, I recommend to the authors to carefully read the text paying attention to le lexicon used and to the terminology. For instance:

LANE 2: what does L. stand for?

LANE 20: the abstract section is not very fluent. I suggest to revise such a section, maybe summarizing results in a more concise way and furnishing a more general evaluation of the data. Before reading the full text, may be little confusing furnishing experimental details, for instance about timing (14 or 21 days).

I suggest some corrections to the text:

LANE 33 and 37: GGT and CRP should be expanded

LANE 40: ‘’hepatal’’, I think that ‘’hepatic’’ sounds better

LANE 76: please revise the sentence, I do not understand why the authors says ‘‘or biological therapy’’.

LANE 79: I think that this sentence is redundant and could be deleted.

LANE 84: ‘’since immemorial’’, please, substitute the term ‘’immemorial’’. For instance, authors may write ‘widely used’.

 I recommend to the authors to carefully read the text paying attention to le lexicon used and to the terminology. 

Therefore, I suggest the publication of the paper.

Author Response

Author's Reply to the Review Report (Reviewer 2) 

Comments and Suggestions for Authors

In the present paper, Chrastina et report a biological characterization of the effect of saffron extract on a model of rheumatoid arthritis. Authors performed lots of assays suggesting interesting properties of such a natural compound on inflammation and arthritis. The rational behind the project design is convincing and experimental data were reasonably  justified by the authors. However, I recommend to the authors to carefully read the text paying attention to le lexicon used and to the terminology. For instance:

LANE 2: what does L. stand for?

Authors´ reply: Thank you for this comment. The “L.” in the botanic is a Latin word and stands for Linnaeus, shortcut “L.” (the taxonomy after Carl von Linné: https://en.wikipedia.org/wiki/Linnaean_taxonomy) .

LANE 20: the abstract section is not very fluent. I suggest to revise such a section, maybe summarizing results in a more concise way and furnishing a more general evaluation of the data. Before reading the full text, may be little confusing furnishing experimental details, for instance about timing (14 or 21 days).

Authors´ reply: Thank you for this comment. We have modified the abstract to be less confusing for the reader in several sentences (please see the abstract). Thank you for this suggestion.

I suggest some corrections to the text:

LANE 33 and 37: GGT and CRP should be expanded

Authors´ reply: We have explained the GGT and CRP in the abstract.

LANE 40: ‘’hepatal’’, I think that ‘’hepatic’’ sounds better

Authors´ reply: We have modified the sentence (please see the abstract).

LANE 76: please revise the sentence, I do not understand why the authors says ‘‘or biological therapy’’.

Authors´ reply: Thank you for this comment. The term “biological therapy” is derived from a nomenclature of disease-modifying antirheumatic drugs. In our humble opinion, and in light of the recent emergence of new therapeutics for rheumatoid arthritis, such as kinase inhibitors and biosimilars, a new nomenclature for disease-modifying antirheumatic drugs (DMARDs), which are currently often classified as synthetic (or chemical) DMARDs (sDMARDS) and biological DMARDs (bDMARDs), has been drafted and it is commonly used in the immunological and rheumatological society (https://ard.bmj.com/content/73/1/3). Moreover, the term “biological medicinal product” and consequently the “biological therapy” by such product has been recognised by the European medicines agency as well (https://www.ema.europa.eu/en/human-regulatory/research-development/scientific-guidelines/biologicals/biologicals-finished-product). We hope, this has elucidate your concerns.

LANE 79: I think that this sentence is redundant and could be deleted.

Authors´ reply: Thank you for this comment. We have rephrase the sentence.

LANE 84: ‘’since immemorial’’, please, substitute the term ‘’immemorial’’. For instance, authors may write ‘widely used’.

Authors´ reply: Thank you for this comment. We have rephrase this word and we hope it will be clearer.

Reviewer 3 Report

Dear authors,

Thank you very much for reviewing your manuscript. I give you the following comment: modify your manuscript to enhance the readability and understanding of your research work for readers.

1.      Please explain more about Crocus sativus L. (saffron - SF) and relation with RA with figure.

2.      Please add schematics of your research work with highlighted novelty of your research works in an introduction section.

3.      Please follow the guidelines of journal to maintain the proper figure and table accordingly.

4.      Please check the check the grammatical errors and crosscheck the reference.

Best Regards

Author Response

Author's Reply to the Review Report (Reviewer 3)

Comments and Suggestions for Authors

Dear authors,

Thank you very much for reviewing your manuscript. I give you the following comment: modify your manuscript to enhance the readability and understanding of your research work for readers.

  1. Please explain more about Crocus sativus L. (saffron - SF) and relation with RA with figure.

Authors´ reply: Thank you for this comment. The relation with human rheumatoid arthritis (RA) cannot be directly accommodated with effects observed in our adjuvant arthritis model. However, some similarities resemble the same pattern, such as local inflammation (joints, spleen, liver, etc.) and systemic (pro-inflammatory cytokines IL1beta, IL-17A, etc.). The effect of saffron on the course of the RA should be studied on humans once the clinical trial is set up. However, the relationship of the results (figures and tables) with saffron extract is described in section "3. Results" in more detail. At the end of the manuscript, we tried to explain the relation of the saffron effect with the course of the disease (AA) briefly in conclusion – section "5. Conclusion). Please see the manuscript's particular sections. We are of the opinion that saffron extract might be a reasonable therapeutic approach for rheumatic patients as it has been drafted with our results in section "3. Results". We hope this has elucidated your concerns and that we have addressed your issue.

  1. Please add schematics of your research work with highlighted novelty of your research works in an introduction section.

Authors´ reply: Thank you for this comment. The authors have prepared the schematics of our research, and it could be visible in section “5. Conclusion”. In our humble opinion, the novelty could also be highlighted at the end of the manuscript. The interpretation of the novelty might be more appropriate in the context of further experiments.

  1. Please follow the guidelines of journal to maintain the proper figure and table accordingly.

Authors´ reply: Thank you for this comment. In our humble opinion, the figures were prepared according to the template (https://www.mdpi.com/files/word-templates/nutrients-template.dot) as well as according to the instructions for authors for Nutrients (https://www.mdpi.com/journal/nutrients/instructions#figures). Moreover, the tables followed the journal´s format. We hope that we have followed the general guideline for the figures and tables in order to be in line with the Nutrients standards. However, the authors are open to discuss the appropriateness of the figures and tables with the Editors of Nutrients as well.

  1. Please check the check the grammatical errors and crosscheck the reference.

Authors´ reply: Thank you for this comment. We have made considerable grammar corrections in the whole text, and in our humble opinion, we hope the grammatical errors were diminished.

Round 2

Reviewer 1 Report

the authors have successfully answered the comments raised by this referee.